# Elevated Cerebrospinal Fluid Proteins and Albumin Determine a Poor Prognosis for Spinal Amyotrophic Lateral Sclerosis

**DOI:** 10.3390/ijms231911063

**Published:** 2022-09-21

**Authors:** Abdelilah Assialioui, Raúl Domínguez, Isidro Ferrer, Pol Andrés-Benito, Mónica Povedano

**Affiliations:** 1Functional Unit of Amyotrophic Lateral Sclerosis (UFELA), Department of Neurology, Bellvitge University Hospital, 08907 L’Hospitalet de Llobregat, Barcelona, Spain; 2Department of Neurology, Consorci Sanitari Alt Penedès-Garraf, 08720 Vilafranca del Penedès, Barcelona, Spain; 3Neuropathology Group, Bellvitge Biomedical Research Institute (IDIBELL), 08907 L’Hospitalet de Llobregat, Barcelona, Spain; 4Neurologic Diseases and Neurogenetics Group, Bellvitge Biomedical Research Institute (IDIBELL), 08907 L’Hospitalet de Llobregat, Barcelona, Spain; 5Biomedical Network Research Center on Neurodegenerative Diseases (CIBERNED), Institute Carlos III, 08907 L’Hospitalet de Llobregat, Barcelona, Spain; 6Department of Pathology and Experimental Therapeutics, University of Barcelona, 08907 L’Hospitalet de Llobregat, Barcelona, Spain

**Keywords:** spinal ALS, CSF, biomarkers, prognostic, albumin

## Abstract

Amyotrophic lateral sclerosis (ALS) is a heterogeneous disease, both in its onset phenotype and in its rate of progression. The aim of this study was to establish whether the dysfunction of the blood–brain barrier (BBB) and blood–spinal cord barrier (BSCB) measured through cerebrospinal fluid (CSF) proteins and the albumin-quotient (QAlb) are related to the speed of disease progression. An amount of 246 patients diagnosed with ALS were included. CSF and serum samples were determined biochemically for different parameters. Survival analysis based on phenotype shows higher probability of death for bulbar phenotype compared to spinal phenotype (*p*-value: 0.0006). For the effect of CSF proteins, data shows an increased risk of death for spinal ALS patients as the value of CSF proteins increases. The same model replicated for CSF albumin yielded similar results. Statistical models determined that the lowest cut-off value for CSF proteins able to differentiate patients with a good prognosis and worse prognosis corresponds to CSF proteins ≥ 0.5 g/L (*p*-value: 0.0189). For the CSF albumin, the QAlb ≥0.65 is associated with elevated probability of death (*p*-value: 0.0073). High levels of QAlb are a bad prognostic indicator for the spinal phenotype, in addition to high CSF proteins levels that also act as a marker of poor prognosis.

## 1. Introduction

Amyotrophic lateral sclerosis (ALS) is a neurodegenerative disease that predominantly affects upper and lower motor neurons, producing focal onset motor deficits. The course of the disease is rapidly progressive, leading to death, usually due to respiratory failure, in three to five years from the onset of symptoms [1]. A large percentage of patients present a purely motor expression; however, others may show associated mild cognitive impairment and altered executive functions, while an even smaller percentage present with a cognitive impairment pattern compatible with frontotemporal dementia (FTD) [2]. About 5–16% of patients have a family history of ALS, FTD or both and are classified as familial ALS (FALS), while the remaining 90% are classified as sporadic ALS (SALS) [3].

ALS is a heterogeneous disease, both in its onset phenotype and in its rate of progression. Because of this, it is very important to find out what factors are associated with a good prognosis as well as with a bad prognosis. In addition to the importance this has for the patient and their environment, since it is often helpful for family planning, the prognosis of the disease at the time of diagnosis is also of great importance for clinical trials [4,5].

Currently, we have sufficient evidence of vascular involvement in ALS as another etiopathogenic mechanism. The alterations described at this level include endothelial cells, tight junctions, pericytes, aquaporin 4, matrix metalloproteinases and cytokines, as well as the expression of VEGF receptors [6,7,8]. The most important and relevant consequence is dysfunction of the blood–brain barrier (BBB) and blood–spinal cord barrier (BSCB) [9]. The objective of the present study was to establish whether the dysfunction of the blood–brain barrier and blood–spinal cord barrier, measured through the CSF proteins and the albumin quotient (QAlb), is related to the speed of disease progression.

## 2. Results

We analyzed 246 patients who were followed up with in the multidisciplinary unit of the Bellvitge University Hospital and who met the El Escorial criteria for a probable or defined category; 43.9% were women and 56.1% men. The mean age at the onset of symptoms was 60.8 (12.4) years. The diagnostic delay from the onset of symptoms was 9.87 [6.00; 13.40] months. A percentage of 70.2% of the patients presented with spinal phenotype, 28.5% bulbar, and 1.24% respiratory; 8.1% of the patients presented pathological expansion (>30 repetitions) in the *C9ORF72* gene (Table 1).

The analytical results for CSF proteins, cholesterol, creatine kinase (CK), CSF albumin, serum albumin and QAlb are detailed in Table 2. In addition, all the patients had brain and cervical MRIs without evidence of structural lesions that could justify the disease and electromyography that showed diffuse involvement of the second motor neuron.

The mean diagnostic delay time for patients with spinal phenotype and elevated CSF protein was 290.6 days (168.76) and 384.65 days (291.84) for patients with normal CSF protein, with a difference approaching statistical significance (*p* = 0.054). We performed a survival analysis based on the phenotype at the time of diagnosis. We observed that patients with bulbar phenotype at the onset of the disease had a higher probability of death compared to patients with spinal onset, with statistically significant differences (Log-rank test, *p*-value: 0.0006) (Figure 1).

In fact, the risk of death at a given time of a patient with a bulbar phenotype is 65.02%, greater than that for a spinal phenotype patient (HR 1.65 (95% CI 1.18 to 2.32) (Table 3).

Next, we analyzed the patient’s death probability from the onset of symptoms based on whether the CK and cholesterol values were high or normal, and we did not find statistically significant differences (*p* = 0.17 and 0.75, respectively). Figure 2 (Left) and Table 4 show the results for CK; Figure 2 (Right) and Table 5 for cholesterol.

Finally, using a raw and an adjusted Cox model for the effect of CSF protein values, we analyzed the effect of CSF protein elevation on patient survival. In the adjusted Cox model, we included the effect of the interaction between the CSF proteins and the phenotype group. Both the crude model and the phenotype-adjusted model showed statistically significant differences for the effect of CSF proteins. For the crude model, if the value of CSF protein increased by a standard deviation (SD) with respect to the mean (increased 0.15 with respect to 0.41), it was expected that the risk of death would be 22% (HR 1.22 (95% CI 1.05 to 1.43)) greater than if there had been no increase given. For the model adjusted for the interaction with the phenotype, in the case of patients with the spinal phenotype, the elevation of the CSF proteins had a significant effect (HR 1.26 (95% CI 1.03 to 1.54)); however, for patients with the bulbar phenotype, with the effect of the interaction at 0.78 (95% CI 0.56 to 1.07), there was no significant effect, with the HR at 0.98 (95% CI 0.58 to 1.65). In other words, if the value of the CSF proteins increased by one SD of the mean (increased 0.15 with respect to 0.41), and if the patient had a spinal phenotype, the increase entailed an increase in the risk of death of 25.66%. In contrast, with the bulbar phenotype, we did not observe a statistically significant effect (Table 6).

In the same manner, we analyzed the effect of elevated CSF albumin on patient survival using a crude and an adjusted Cox model and obtained results similar to those previously obtained using CSF proteins. For the crude model, if the CSF albumin value increased one standard deviation (SD) with respect to the mean (increased 118.2 with respect to 273.45), it could be expected that the risk of death would increase by 27.7% (HR 1.28 (95% CI 1.04 to 1.57).

For the model adjusted for the interaction with the phenotype, in the case of patients with the spinal phenotype, the elevation of the CSF albumin had a significant effect of HR 1.66 (95% CI 1.22 to 2.26). However, for patients with the bulbar phenotype, with the effect of the interaction being 0.63 (95% CI 0.41 to 0.96), it did not have a significant effect, with the HR being 1.04 (95% CI 0.50 to 2.17). In other words, if the CSF albumin value increased by one SD compared to the mean (117.26 compared to 273.77), and if the patient had a spinal phenotype, this increase meant an increase in the risk of death of 66.12%. In contrast, if the patient had a bulbar phenotype, we did not observe a statistically significant effect (Table 7).

Given that both are continuous quantitative variables, once we confirmed the negative effect on patient survival of the increase in both CSF proteins and albumin, we considered the possibility of seeking the lowest cut-off value that could differentiate patients with a good vital prognosis from those with a worse prognosis. In this way, we could analyze the patient’s death probability in relation to CSF protein values. In the survival curves, according to whether the CSF protein level was ≥0.46 g/L, we observed an increased death probability for the group with elevated CSF protein, but this difference was not statistically significant (Log-rank test, *p*-value = 0.1282). Given this, in the model used previously in which the patient’s death probability was directly proportional to the increase in CSF proteins, we successively increased the cut-off point until we reached statistically significant differences with the CSF protein cut-off value> 0.5 g/L; Log-rank test, *p*-value: 0.0189 (Figure 3 Left).

Finally, to find the cut-off value for albumin, we considered that the QAlb index would be more sensitive and specific than the absolute value for albumin. We calculated the QAlb and created two survival curves, one corresponding to patients with a normal index <0.65 and the other to patients with an index considered elevated at ≥0.65. In the two curves, we observed a worse vital prognosis for patients with a high index when compared to the normal index along with statistically significant differences; Log-rank test, *p*-value: 0.0073 (Figure 3 Right).

## 3. Discussion

This work represents the first observational study to correlate CSF proteins and albumin elevation with the rate of the progression of the spinal phenotype of ALS. Using routine hospital biochemical parameter analysis, we have reported additional biomarkers in CSF to complement the current biochemical tools in ALS-patient-prognosis management. It is well-known that ALS is a heterogeneous disease in its initial clinical manifestations, as well as in its evolutionary progression [10]. Recently, numerous epidemiological (age of onset, sex), genetic and biological poor prognostic factors have been described [4,11,12,13]. In the present study, we have tried to evaluate CSF protein, albumin, and QAlb index elevation as poor prognostic factors. CSF protein and albumin elevation has already been described in several studies as a finding in the CSF of ALS patients, but it had not been evaluated as a marker of poor prognosis [14,15] nor on the basis of modern ALS diagnosis criteria and onset classification [16]. However, a previous study highlighted the use of the CSF total protein parameter as a potential survival biomarker in a homogeneous group of familial ALS patients with non-defined mutation [16]. In addition, we also studied creatine kinase (CK) and cholesterol levels due to their relationship with the disease, despite their easy detection as routine analytes in hospital laboratories and previous controversial findings related to them. Since CK has been associated with muscle damage, Creatine kinase (CK) levels have been studied in the context of ALS to explore their relationship to the clinical characteristics and survival prognoses of ALS patients [17,18]. Abnormal levels of cholesterol and other lipids in the circulatory system and CNS have been reported to be higher in ALS patients than in controls, making them part of the multiple pathogenic mechanisms that have been proposed as contributing to ALS, but also without a consensus [19]. However, our results do not relate altered levels in these parameters to any other clinical parameter within our cohort, or to QAlb and CSF total protein levels.

Given that albumin is a protein of pure liver synthesis [20], its elevation in the CSF is indicative of the dysfunction of the blood–brain barrier (BBB) [21,22]. In addition to the CSF analytical findings, BBB dysfunction was demonstrated using pathological studies of autopsy specimens from ALS patients. Winkler et al. analyzed 11 postmortem tissue samples and found signs compatible with blood–brain barrier rupture [23]. Since albumin is the most abundant protein substance in the CSF [24], elevation of the proteins is indicative of albumin elevation and therefore BBB dysfunction. However, QAlb is considered a more specific marker of BBB dysfunction because of the direct correlation between the CSF albumin concentration and the serum albumin concentration [25]. In addition, due to this last fact, it may even be more sensitive, given the high frequency of protein malnutrition in the disease [26]. CSF proteins, which represent a continuous quantitative variable, pose a great difficulty in establishing the maximum value of normality. It is accepted that a protein level ≥ 0.46 g/L is altered [27], but recent studies have shown that this value should not be considered altered for all ages [28]. In any case, the objective of our study was not to study the upper and lower limits of normality for CSF proteins and/or albumin, but rather to find out if the elevation of these parameters would lead to a worse evolutionary prognosis and also, to attempt to establish a cut-off value with clinical utility for the prediction of the evolution of the disease at the time of diagnosis. Unlike other prognostic biomarkers, CSF proteins and albumin are easy parameters to obtain, well within the reach of practically all clinical analysis laboratories.

In our study, the values for CSF proteins and the QAlb in relation to a poor prognosis for the disease in the spinal phenotype were ≥0.5 g/L and ≥0.65, respectively. It should be noted that for patients with the spinal phenotype, the diagnostic delay time for those with elevated CSF proteins was shorter compared to the time for patients with normal CSF proteins, with a difference approaching statistical significance, and this suggests that the speed of disease progression is different in the two groups from the onset of the disease.

Regarding the bulbar phenotype, we were unable to find statistically significant or clinically relevant differences between patients with elevated CSF protein or QAlb and those with normal parameters for them. This was probably due to the effect of the bulbar phenotype on the BBB and on BSCB dysfunction being different from that occurring in spinal phenotype. The BBB is a highly selective, semipermeable complex that surrounds most of the blood vessels in the brain. Maintenance of the BBB is essential for tight control of the chemical composition of the brain’s interstitial fluid (ISF), essential for synaptic function as well as for offering a form of protection against bloodborne pathogens [29]. However, it is not a homogenous structure, showing different regional compositions, such as those observed in the circumventricular organs (CVOs) centered around the ventricles of the brain, and [30,31] also showing different blood–CNS permeability capabilities in a region-dependent manner [32,33,34]. Similarly, separation of the central nervous system (CNS) from the cardiovascular system occurs via the blood–spinal cord barrier (BSCB). The BSCB acts as a functional equivalent for the BBB by maintaining the microenvironment for the cellular constituents of the spinal cord. Even if the BSCB could intuitively be considered as the morphological extension of the BBB into the spinal cord, evidence suggests that this is not the case. The BSCB shares the same principal building blocks with the BBB; nevertheless, it seems that morphological and functional differences may exist between these two, suggesting that the BSCB is more permeable [35]; it also shows functional differences in a region/area-dependent manner [36,37]. Thus, structural and functional differences between the BBB and the BSCB along with changes in structure and function throughout each structure in a region-dependent way may explain how the kinds of involvement could be different in the context of each ALS onset. In addition to this idea, a recent study also points out different degrees of BBB/BSCB involvement among stratified ALS patients based on their clinical onset and symptoms, such as the presence or absence of FTD [38]. Therefore, patients with the bulbar phenotype worsen rapidly, but they do not manifest the same effects on the BBB and BSCB dysfunction parameters measured in our study and altered in the spinal onset ALS cohort showing the elevation of CSF proteins and QAlb. These results are reinforced by Prell et al., who also found an association with increased levels of albumin in CSF, spinal-onset disease and BBB damage [39].

In Figure 1 and Table 3, we show the probability of death for patients with the bulbar phenotype compared to the spinal one. The greater risk of death for the bulbar phenotype compared to the spinal one during the evolution of the disease is notable, reaching 65%. Therefore, we think that the risk of death associated with the bulbar phenotype is much higher than for any other poor prognostic factor, and it may mask other factors.

From a clinical point of view, we clearly see that a patient with a bulbar phenotype has a poor prognosis for the disease [40], but it is not easy to determine the prognosis of a patient with a spinal phenotype. Therefore, the association of CSF protein elevation with the speed of disease progression can be of great help in this regard. In physiological conditions, the BBB and BSCB avoid the penetrance to the central nervous system of leukocytes, erythrocytes and other circulating molecules [41,42]. Several studies have provided data about peripheral blood immune activation in ALS [43]. In the context of patients with the spinal onset of ALS, the increase of CSF total proteins and QAlb may be related with the loss of BBB integrity that allows infiltration into the CNS of the cytokines, chemokines, and other proinflammatory mediators, and of peripheral leukocytes such as monocytes, macrophages and CD4 T cells that may interact with activated astrocytes and microglia, leading to accelerated disease progression [41,43,44,45,46]. In fact, in a mutant SOD1 mouse model and also in rat models bearing the same mutation, the BSCB damage occurred before motor neuron degeneration, indicating that it was implicated in the initiation of the disease [47,48,49]. In our study, we cannot be certain at which stage of the disease the dysfunction of the blood–brain barrier is involved. However, our data suggest that blood–brain barrier dysfunction accelerates disease progression from the onset of symptoms and decreases spinal phenotype patient survival.

## 4. Materials and Methods

Patients’ inclusion and clinical classification criteria: Medical records of patients who were diagnosed with ALS and who were undergoing quarterly check-ups in the multidisciplinary unit of motor neuron diseases at the Bellvitge University Hospital were reviewed. These patients were ambispectively followed. Later, patients with a diagnosis given within the period between 2012 and 2019 were included; they were followed up with prospectively from 2012 to 2022. All the patients included met the El Escorial criteria for a definite and/or probable category [50,51]. Patients with other motor neuron diseases were excluded, including patients with concomitant polyneuropathies or myelopathy, recent ischemic or hemorrhagic stroke, multifocal motor neuropathy, chronic inflammatory demyelinating polyneuropathy, or neuroborreliosis. During the study, 18 patients were excluded, and 8 patients dropped out. Finally, 246 patients were included, of whom 163 died during the follow-up period. All patients underwent a systematic first visit in outpatient clinics that included a neurological examination, followed by a cranial and cervical MRI and subsequent referral to the diagnostic protocol that included electroneurography, electromyography, blood analysis, lumbar puncture and in some cases, transcranial magnetic stimulation. Subsequently, the patients were given results at a visit in which they were informed of the diagnosis of a neurodegenerative disease of the motor neurons. Finally, the patients were scheduled for routine quarterly check-ups in a multidisciplinary unit composed of neurology, pneumology, endocrinology, nutrition, rehabilitation and nursing.

ALS patients were evaluated clinically according to the main signs at onset (spinal, bulbar and respiratory). We define spinal onset ALS as when the weakness starts in the lower or upper limbs. Clinical examination must show atrophy, weakness of muscles, fasciculations and hyperreflexia. In bulbar onset ALS, the disease starts with dysarthria, dysphagia and sometimes, tongue fasciculations. Disease spread involving the limbs and clinical examinations reveals the presence of limb hyperreflexia. In respiratory onset ALS, the patients begin with orthopnea or dyspnea, sometimes accompanied by mild spinal or bulbar signs. All the diagnoses were confirmed by electromyography and the monitoring of their evolution.

Clinical variables: We calculated the diagnostic delay as the diagnosis date minus the start date, the start age as the start date minus the date of birth and the disease duration as the death date minus the start date.

Cerebrospinal fluid analysis: Lumbar puncture (LP) was performed by neurologists in outpatient clinics, as LP is performed routinely in all patients with suspected motor neuron diseases to rule out other diseases of the peripheral nervous system. LP was performed in parallel to serum and plasma extraction and analysis in all patients after they had given informed consent. CSF was prospectively collected from patients undergoing lumbar puncture due to the clinical suspicion of motor neuron disease at the functional unit of amyotrophic lateral sclerosis (UFELA) of the Neurology Service of the Bellvitge University Hospital at the time of diagnosis. A routine analysis of CSF did not detect the presence of red blood cells in any case. In these patients, 1.5 ± 0.5 mL of CSF was collected in polypropylene tubes. The CSF was centrifuged at 3000 rpm for 15 min at room temperature. The supernatant was collected and aliquoted in volumes of 220 μL, and it was either used immediately for analysis in the laboratory of the same hospital or stored at −80 °C for future use. Results included glucose, proteins, cells, and albumin in cerebrospinal fluid (CSF), as well as cholesterol, creatine kinase, and albumin in blood serum. The albumin quotient (QAlb) was calculated with the formula QAlb = CSF albumin/serum albumin. The abnormality value for CSF proteins was established at ≥ 0.46 g/L according to the parameters of the clinical analysis laboratory of the Bellvitge Hospital [23].

Genetic analysis: DNA extraction was performed through peripheral blood obtained by venipuncture. The hexanucleotide expansion in *C9ORF72* was analyzed with PCR with subsequent fragment analysis and repeat primed-PCR. GGGGCC hexanucleotide with a repeat length of > 30 was used as the standard threshold for distinguishing between neutral and pathogenic expansions [52].

Statistical analysis: The demographic and clinical profiles of included subjects were described by counts and percentages for categorical variables and the mean with standard deviation or the median with the first and last quartile for numerical variables. Cumulative incidence of death per 100 subjects and incidence per 1000 subjects a year were estimated using the onset symptoms as a baseline time. The Kaplan–Meier estimator of the probability of death by time was represented. Raw and adjusted hazard ratios (HR) were used to assess the association between death and a number of risk factors, such as phenotype, creatine-kinase (CK), cholesterol, albumin, and CSF protein levels. HR was estimated using a Cox regression model. Adjusting variables were age, sex and *C9ORF72*. Age was introduced in the model as a time-dependent covariate, considering age at the time of onset and at the time of diagnosis. The application conditions of the models were validated and confidence intervals at 95% of the estimators were calculated whenever possible. All analyses were carried out with the statistical package R version 4.0.5 (2021-03-31) for Windows.

## 5. Conclusions

The best prognostic indicator of the spinal phenotype is QAlb. In the absence of QAlb, a level of CSF proteins ≥ 0.5 g/L is a clear marker of poor prognosis in the spinal phenotype. Our data suggest that blood–brain barrier dysfunction does not increase as the disease progresses, because patients with normal proteins had the longest diagnostic delay, and therefore, CSF collection was performed at a more advanced stage of the disease.

## Figures and Tables

**Figure 1 ijms-23-11063-f001:**
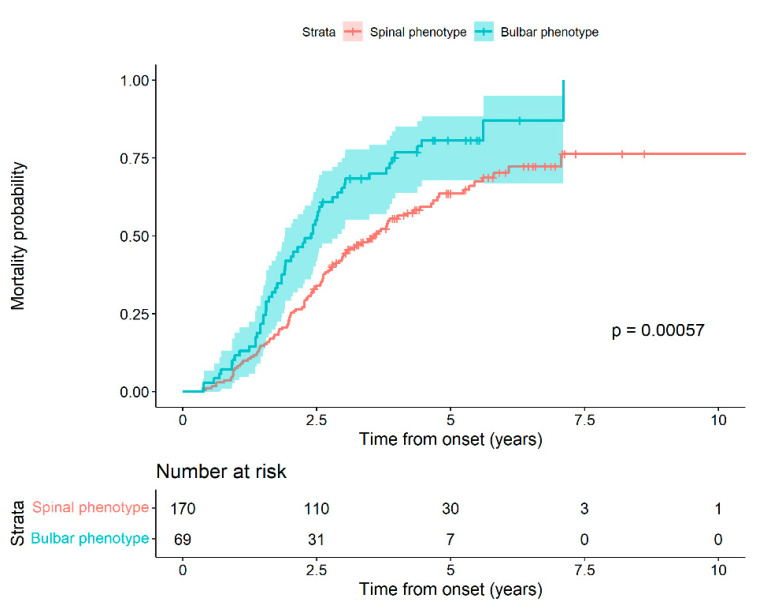
Mortality probability in relation to phenotype as a function of follow-up time from onset. The differences observed in the two survival curves allow us to reject the null hypothesis of equality (Log-rank test, *p*-value: 0.0006).

**Figure 2 ijms-23-11063-f002:**
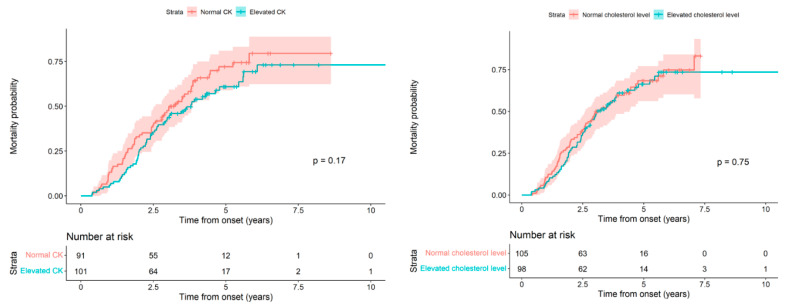
(**Left**) Mortality probability as a function of the follow-up time from onset, comparing levels of CK. The differences observed in the two survival curves according to the CK level do not allow us to reject the null hypothesis of equality (Log-rank test, *p*-value: 0.1731). (**Right**) Mortality probability as a function of the follow-up time from onset, comparing cholesterol levels. The differences observed in the two survival curves according to cholesterol level do not allow us to reject the null hypothesis of equality (Log-rank test, *p*-value: 0.7478).

**Figure 3 ijms-23-11063-f003:**
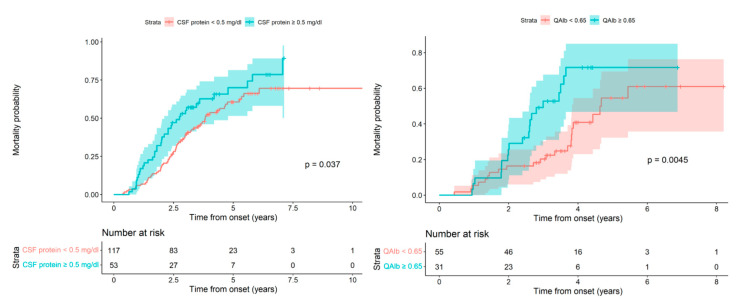
(**Left**) Mortality probability as a function of the follow-up time from onset when comparing levels of CSF proteins using 0.5 g/L as a cut-off point for those with a spinal phenotype. We observed an increased probability of death in patients with CSF protein ≥ 0.5 g/L when compared to those with CSF protein < 0.5 g/L (**Right**). Mortality probability as a function of the follow-up time from onset comparing the two QAlb levels for the spinal phenotype. We observed an increased probability of death in patients with QAlb ≥ 0.65 compared to those with QAlb <0.65.

**Table 1 ijms-23-11063-t001:** Clinical features of ALS patients.

Clinical Feature	Cases (*n* = 246)	
Gender, N (%):		246
Women	108 (43.9%)	
Man	138 (56.1%)	
Age at onset, Mean (SD)	60.8 (12.4)	244
Diagnostic delay (months), Median [Q1;Q3]	9.87 [6.00;13.4]	246
Phenotype, N (%):		242
Spinal	170 (70.2%)	
Bulbar	69 (28.5%)	
Respiratory	3 (1.24%)	
*C9ORF72*, N (%):		246
Normal	226 (91.9%)	
Expanded	20 (8.13%)	

**Table 2 ijms-23-11063-t002:** Biochemical analysis data for CSF and plasma from ALS patients.

Biochemical Parameters		Cases (*n* = 246)	
CSF protein, Mean (SD)	(≥0.46 g/L)	0.41 g/L (0.15)	246
CSF protein, Median [Q1;Q3]		0.38 g/L [0.31;0.49]	246
CSF protein level, N (%):			246
Normal		173 (70.3%)	
Elevated		73 (29.7%)	
CK (plasma), Median [Q1;Q3]	(>2.48 uKat/L)	3.16 uKat/L [2.04;5.41]	192
CK level (plasma), N (%):			192
Normal		91 (47.4%)	
Elevated		101 (52.6%)	
Cholesterol (plasma), Mean (SD)	(>5.18 mmol/L)	5.14 mmol/L (1.08)	203
Cholesterol level (plasma), N (%):			203
Normal		105 (51.7%)	
Anormal		48 (39.0%)	
CSF albumin, Mean (SD)	(>350 mg/L)	273 (118)	137
CSF albumin, Median [Q1;Q3]		241 [198;330]	137
Serum albumin, Median [Q1;Q3]		45,000 [43,000;47,000]	168
QAlb, N (%):	(≥ 0.65)		123
Normal		75 (61.0%)	
Abnormal		48 (39.0%)	

**Table 3 ijms-23-11063-t003:** Adjusted Cox model for phenotype. For this model, given two subjects with different phenotypes alive at a determined time, the risk of death at any time thereafter for the patient with the bulbar phenotype is expected to be 65% higher than that for the spinal phenotype (HR 1.65). Age corrected for time refers to the time elapsed between onset and diagnosis.

Adjusted Cox Model				
Predictors	Estimates	Std. Error	CI	*p*-Value
Bulbar	1.65	0.29	1.18–2.32	0.004
Man	1.43	0.24	1.03–2.00	0.035
*C9ORF72*	1.28	0.40	0.69–2.37	0.434
Age corrected for time	1.03	0.01	1.02–1.04	<0.001
Observations	473			
R2 Nagelkerke	0.065			
AIC	1532.124			

**Table 4 ijms-23-11063-t004:** Adjusted Cox model based on CK values. The model does not allow us to conclude that for a subject alive at a determined time, if the CK is higher, the risk of death at any time thereafter increases. Age corrected for time refers to the time elapsed between onset and diagnosis.

	Adjusted Cox Model			
Predictors	Estimates	Std. Error	CI	*p*-Value
Scale(Ck)	0.99	0.11	0.80–1.24	0.961
Man	1.49	0.30	1.01–2.21	0.046
*C9ORF72*	1.15	0.43	0.56–2.37	0.701
Age corrected for time	1.03	0.01	1.02–1.05	<0.001
Bulbar phenotype	1.52	0.32	1.01–2.31	0.047
CK-phenotype interaction	0.74	0.19	0.45–1.23	0.244
Observations	376			
R2 Nagelkerke	0.083			
AIC	1090.810			

**Table 5 ijms-23-11063-t005:** Adjusted Cox model according to cholesterol value. The model does not allow us to conclude that for a subject alive at a determined time, if cholesterol is higher, the risk of death at any time thereafter increases. Age corrected for time refers to the time elapsed between onset and diagnosis.

	Adjusted Cox Model			
Predictors	Estimates	Std. Error	CI	*p*-value
Scale(cholesterol)	0.90	0.08	0.75–1.07	0.219
Man	1.46	0.27	1.01–2.11	0.044
*C9ORF72*	1.12	0.43	0.53–2.37	0.770
Age corrected for time	1.03	0.01	1.01–1.05	<0.001
Bulbar phenotype	1.69	0.32	1.16–2.46	0.007
Observations	395			
R2 Nagelkerke	0.073			
AIC	1187.837			

**Table 6 ijms-23-11063-t006:** Adjusted Cox model according to CSF protein values. In this model that is adjusted for the interaction with the phenotype, the HR of the interaction is 0.78. Therefore, for patients with the bulbar phenotype, the HR of CSF protein is 0.98 (95% CI 0.58 to 1.65) while for those with the spinal phenotype, it is 1.26. This means that—given a subject alive at a determined time for whom we increase the CSF protein value by one SD with respect to the mean (0.15 with respect to 0.41)—if the patient has a spinal phenotype, the increase will mean an increase in the risk of death of 25.66%. However, when the patient is of the bulbar phenotype, we do not observe a statistically significant effect. Age corrected for time refers to the time elapsed between onset and diagnosis.

	Adjusted Cox Model			
Predictors	Estimates	Std. Error	CI	*p*-Value
Scale (CSF proteins)	1.26	0.13	1.03–1.54	0.028
Man	1.32	0.24	0.93–1.87	0.124
*C9ORF72*	1.25	0.40	0.68–2.33	0.473
Age corrected for time	1.03	0.01	1.01–1.04	<0.001
Bulbar phenotype	1.66	0.29	1.18–2.33	0.004
CSF proteins-phenotype interaction	0.78	0.13	0.56–1.07	0.123
Observations	473			
R2 Nagelkerke	0.075			
AIC	1531.446			

**Table 7 ijms-23-11063-t007:** Adjusted Cox model according to the CSF albumin values. In this model adjusted for the interaction with the phenotype, the HR of the interaction is 0.63. Therefore, for patients with the bulbar phenotype, the HR of CSF albumin is 1.04 (95% CI 0.50 to 2.17) while for those with the spinal phenotype, it is 1.66. This means that if the CSF albumin value is increased by one SD compared to the mean (117.26 compared to 273.77), and if the patient has a spinal phenotype, this increase will mean an increase in the risk of death of 66.12%. In contrast, if the patient has a bulbar phenotype, we do not observe a statistically significant effect. Age corrected for time refers to the time elapsed between onset and diagnosis.

	Adjusted Cox Model			
Predictors	Estimates	Std. Error	CI	*p*-Value
Scale (CSF albumin)	1.66	0.26	1.22–2.26	0.001
Man	1.19	0.31	0.71–1.99	0.505
*C9ORF72*	1.25	0.92	0.30–5.27	0.763
Age corrected for time	1.04	0.01	1.02–1.07	<0.001
Bulbar phenotype	1.88	0.47	1.15–3.08	0.011
CSF albumin-phenotype interaction	0.63	0.14	0.41–0.96	0.030
Observations	258			
R2 Nagelkerke	0.128			
AIC	609.918			

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
