# Peer review of "Elevated Cerebrospinal Fluid Proteins and Albumin Determine a Poor Prognosis for Spinal Amyotrophic Lateral Sclerosis"

_ijms, 2022, doi:10.3390/ijms231911063_

Round 1
Reviewer 1 Report
The authors are presenting the results of CSF analysis in 246 ALS patients. The conclusion is that “the best prognostic 32 indicator for the spinal phenotype is QAlb, in addition to high CSF-proteins level as a clear marker 33 of poor prognosis”
This work is well done and methodology is clear.
My main concern is that such a work is not new! In 1993 Forbes Norris, with a larger sample (n=385), presented conclusions of the same kind. This author is not even quoted in this manuscript. Additionally, many papers have dealt with CSF in ALS searching for prognostic markers. To date, the best prognosis marker in the neurofilament light chain.
But this work is about a very general analysis of the CSF, while a series of proteins have demonstrated their interest. Why being so general?
While well done and presented, this work does not add significant data to the knowledge on prognosis factors in ALS nor about the role of the BBB.
Author Response
Dear reviewer,
Thank you for your comments and suggestions that allowed us to greatly improve the quality of our manuscript. We agree with all your comments, and we corrected point by point accordingly. The introduced changes are indicated in green in the manuscript and we answered each comment in the following lines.
The authors are presenting the results of CSF analysis in 246 ALS patients. The conclusion is that “the best prognostic indicator for the spinal phenotype is QAlb, in addition to high CSF-proteins level as a clear marker of poor prognosis”.
A: Abstract conclusion has been rewritten.
This work is well done and methodology is clear.
My main concern is that such a work is not new! In 1993 Forbes Norris, with a larger sample (n=385), presented conclusions of the same kind. This author is not even quoted in this manuscript.
A: We included Forbes 1993 quotation in the draft. However, it is important to note that comparison between studies is not possible because the clinical diagnosis determination and the ALS onset criteria do not correspond with those used currently worldwide, which are nowadays more accurate and able to stratify patient clinics. However, we make note of Forbes’ results about the potential of total CSF proteins as a survival biomarker in a more homogeneous and stratified group of familial ALS cases.
Additionally, many papers have dealt with CSF in ALS searching for prognostic markers. To date, the best prognosis marker in the neurofilament light chain. But this work is about a very general analysis of the CSF, while a series of proteins have demonstrated their interest. Why being so general?
A: The results of the present study are intended to support current clinical parameters and already studied biomarkers in the prognosis of ALS disease, using the detection of albumin and CSF total proteins levels, which are routine detection analytes in hospital laboratories.
While well done and presented, this work does not add significant data to the knowledge on prognosis factors in ALS nor about the role of the BBB.
A: Using an easily analyzable biochemical parameter, we have reported additional biomarkers to complement the current biochemical biomarkers and other clinical tools in ALS patient prognosis management.
Reviewer 2 Report
This paper is not of great novelty. It was already shown that the CSF markers are relevant for the spinal offset. However, it remains a challenge to explain the lack of these markers for the bulbar form. The authors discuss this by hypothesizing that in the latter the BBB is not affected early enough (line 247-249). This is not easy to grasp since the BBB breach is considered an early marker in ALS (in addition to cited Zhong et al. Nature Neuroscience, 2008, 11, 420–422; also see e.g. Andjus et al. Anat Rec (Hoboken). 2009;292:1882-1892; Bataveljić et al. Gen Physiol Biophys. 2009;28 Spec No:212-218) and even the authors acknowledge that the BBB is hampered in the bulbar form but not early enough, however this is not in concordance with BBB breach being an early marker. There is only a mention of the BSCB in the end of the Discussion, however a further elaboration on this pathway as well as on the Choroid plexus has also to be taken into account.
It is of course very simplistic to call upon only one parameter such as the QAb as a prognostic marker in ALS that is consider a multimodal disease without a straightforward link to BBB breach (e.g. see Prell et al. Front Neurosci. 2021;15:656456). This has to be discussed.
In addition, it is not clear what is the rationale to include also CK and cholesterol in the study.
The Materials and Methods are written in general terms without real technical details necessary for reproducing the research.
Author Response
Dear reviewer,
First of all, thank you for your comments and suggestions that allowed us to greatly improve the quality of the manuscript. We agree with all your comments, and we corrected point by point the manuscript accordingly. The introduced changes are indicated in green in the manuscript and we answered each comment in the following lines.
This paper is not of great novelty. It was already shown that the CSF markers are relevant for the spinal offset. However, it remains a challenge to explain the lack of these markers for the bulbar form. The authors discuss this by hypothesizing that in the latter the BBB is not affected early enough (line 247-249). This is not easy to grasp since the BBB breach is considered an early marker in ALS (in addition to cited Zhong et al. Nature Neuroscience, 2008, 11, 420–422; also see e.g. Andjus et al. Anat Rec (Hoboken). 2009;292:1882-1892; Bataveljić et al. Gen Physiol Biophys. 2009;28 Spec No:212-218) and even the authors acknowledge that the BBB is hampered in the bulbar form but not early enough, however this is not in concordance with BBB breach being an early marker. There is only a mention of the BSCB in the end of the Discussion, however a further elaboration on this pathway as well as on the Choroid plexus has also to be taken into account.
A: We have included these new citations and reformulated the discussion about differences in results between types of ALS onset, pointing out that these differences may be related to BBB and BSCB involvement, and taking in consideration that these are structures with differences in composition and function between them.
It is of course very simplistic to call upon only one parameter such as the QAb as a prognostic marker in ALS that is consider a multimodal disease without a straightforward link to BBB breach (e.g. see Prell et al. Front Neurosci. 2021;15:656456). This has to be discussed.
A: We referenced Prell et al., and maintained that their results are in partial agreement with ours. The results reported by Prell et al. highlight the clinical heterogeneity of ALS but relate limb-onset disease with high albumin levels in CSF and associate these with BBB damage.
In addition, it is not clear what is the rationale to include also CK and cholesterol in the study.
A: We included them as biochemical parameters studied over time in the context of ALS, despite the lack of consensus, with the aim of associating them with possible changes in QAlb and CSF total protein levels. We have included citations and justification about this in the discussion.
The Materials and Methods are written in general terms without real technical details necessary for reproducing the research.
A: We have expanded this section, specifying the methodology.
Reviewer 3 Report
Assialioui A, et al. reported elevated CSF-protein, CSF-albumin, and QAlb to be prognostic markers among ALS patients. It was established for spinal cord phenotype but not for bulbar type. They also discussed relationship between protein transmission across the blood-brain-barrier and ALS pathogenesis. The study is well designed and potentially has clinical importance. The text and display are concise. Sample size is enough large to make conclusion. However, I have some points to improve readability of the manuscript.
Major points:
1) It is a prospective study. The material and method section has to clarify the period of patient registration(for example, 2010-2015), follow-up period (for example, 2010-2020), drop-out rate, exclusion criteria (for example, comorbidity of neurological disorders that can influence CSF protein), number of excluded subjects, and overall prevalence (or N) of death during the follow-up period.
2) Please clarify definition and inclusion criteria of phenotypes of ALS, including bulbar, spinal cord, and respiratory ones, in the material and method section.
3) I recommend assessing hazard ratio of QAlb for survival in the same ways as tables 6 (CSF protein) and 7 (CSF albumin). It is of importance when we discuss peripheral protein transmission into CSF in ALS pathogenesis.
Minor points:
1) In the table 2: Please provide units (mg/dL, U/dL, or any) of CSF protein, CK, and cholesterol.
2) In the table 2: Please display cut-off value of elevated CSF-protein, elevated CK, and elevated cholesterol.
3) In the tables 3-7: What does ‘Surviv(tstart, tstop, EVENT)’ mean? HR toward death? Please make more concise display here.
4) In the tables 3-7: What does ‘Age corrected for time’ mean? Age at disease onset, age at CSF examination, or age at death? Please make an explanation in table captions.
5) In the tables 3-7: What does ‘risk of death at time t+1’ in captions mean? What is the definition of ‘t+1’? I recommend make an easier writing here, such as cumulative risk (prevalence) of death during study period.
6) Please clearly state that serum albumin was sampled at the same time as sampling of CSF but not in other days.
7) In the page 7: Please clarify whether QAlb was available from all study subjects in the results section.
8) Page2: A literature citation is required for the El Escorial criteria.
9) Page 2: A literature citation is required for cut-off value of C9ORF72 gene hexanucleotide expansion.
Author Response
Dear reviewer,
First of all, thank you for your comments and suggestions that allowed us to greatly improve the quality of the manuscript. We agree with all your comments, and we corrected point by point the manuscript accordingly. The introduced changes are indicated in green in the manuscript and we answered each comment in the following lines.
Assialioui A, et al. reported elevated CSF-protein, CSF-albumin, and QAlb to be prognostic markers among ALS patients. It was established for spinal cord phenotype but not for bulbar type. They also discussed relationship between protein transmission across the blood-brain-barrier and ALS pathogenesis. The study is well designed and potentially has clinical importance. The text and display are concise. Sample size is enough large to make conclusion. However, I have some points to improve readability of the manuscript.
Major points:
1) It is a prospective study. The material and method section has to clarify the period of patient registration(for example, 2010-2015), follow-up period (for example, 2010-2020), drop-out rate, exclusion criteria (for example, comorbidity of neurological disorders that can influence CSF protein), number of excluded subjects, and overall prevalence (or N) of death during the follow-up period.
A: This has been included in MandM, in the section on patient inclusion and clinical classification criteria.
2) Please clarify definition and inclusion criteria of phenotypes of ALS, including bulbar, spinal cord, and respiratory ones, in the material and method section.
A: This has been included in MandM, in the section on patient inclusion and clinical classification criteria.
3) I recommend assessing hazard ratio of QAlb for survival in the same ways as tables 6 (CSF protein) and 7 (CSF albumin). It is of importance when we discuss peripheral protein transmission into CSF in ALS pathogenesis.
A: Given that we have confirmed that the elevation of CSF albumin causes an increased risk of death in patients with spinal ALS, we think that looking for this same effect using QAlb does not add any conceptual value to the study because it is the same effect that causes the elevation of albumin. Moreover, the main objective of the study was to find parameters that can identify those patients with a poor prognosis, and QAlb ≥0.65 demonstrates this perfectly.
Minor points:
1) In the table 2: Please provide units (mg/dL, U/dL, or any) of CSF protein, CK, and cholesterol.
A: Data have been included in Table 2.
2) In the table 2: Please display cut-off value of elevated CSF-protein, elevated CK, and elevated cholesterol.
A: Data have been included in Table 2.
3) In the tables 3-7: What does ‘Surviv(tstart, tstop, EVENT)’ mean? HR toward death? Please make more concise display here.
A: The table titles were shown as the R code instruction of the model, including the adjusted variables (start time, stop time, and the event (death)). We replaced this with “Adjusted Cox model”, which defines the model and provides an estimate of the survival after adjustment for other explanatory variables.
4) In the tables 3-7: What does ‘Age corrected for time’ mean? Age at disease onset, age at CSF examination, or age at death? Please make an explanation in table captions.
A: Age corrected for time refers to the time elapsed between onset and diagnosis. We included this information in table captions.
5) In the tables 3-7: What does ‘risk of death at time t+1’ in captions mean? What is the definition of ‘t+1’? I recommend make an easier writing here, such as cumulative risk (prevalence) of death during study period.
A: We have modified the text to make it more understandable. In these models, "t" refers to a determined time and "t+1" to any time thereafter.
6) Please clearly state that serum albumin was sampled at the same time as sampling of CSF but not in other days. A:
A: LP is performed in parallel to serum and plasma extractions and analysis in all patients after having given informed consent. This has been included in MandM, in the CSF analysis section.
7) In the page 7: Please clarify whether QAlb was available from all study subjects in the results section.
A: When we were mounting the template format, we omitted this information in Table 2. We have included it again and indicated the number of cases included.
8) Page 2: A literature citation is required for the El Escorial criteria.
A: This has been included in in MandM, in the section on patient inclusion and clinical classification criteria, citing: “B.R., Miller R.G., Swash M., Munsat T.L. El Escorial revisited: Revised criteria for the diagnosis of amyotrophic lateral sclerosis. Amyotroph. Lateral Scler. 2000;1:293–299”; and “De Carvalho M., Swash M. Awaji diagnostic algorithm increases sensitivity of El Es-corial criteria for ALS diagnosis. Amyotroph. Lateral Scler. 2009;10:53–57”.
9) Page 2: A literature citation is required for cut-off value of C9ORF72 gene hexanucleotide expansion.
A: This has been included in MandM, in the genetic analysis section, citing “DeJesus-Hernandez M, Mackenzie IR, Boeve BF, Boxer AL, Baker M, Rutherford, NJ, Ni-cholson AM, Finch NA, Flynn H, Adamson J (2011) Expanded GGGGCC hexanucleotide repeat in noncoding region of C9ORF72 causes chromosome 9p-linked FTD and ALS. Neuron 72:245–256”.